# Comparison of Industrial Quenching Oils

**Jiří Hájek** [1],*[ID]**, Zaneta Dlouha** [2] **and Vojtěch Průcha** [2][ID]

1 COMTES FHT a. s., Prumyslova 995, 334-41 Dobrany, Czech Republic
2 Faculty of Mechanical Engineering, University of West Bohemia, Univerzitni 8, 301-00 Plzen, Czech Republic; dlouha@students.zcu.cz (Z.D.); vprucha@kmm.zcu.cz (V.P.)
* Correspondence: jiri.hajek@comtesfht.cz; Tel.: +42-037-719-7333

**Abstract:** This article is a response to the state of the art in monitoring the cooling capacity of quenching oils in industrial practice. Very often, a hardening shop requires a report with data on the cooling process for a particular quenching oil. However, the interpretation of the data can be rather difficult. The main goal of our work was to compare various criteria used for evaluating quenching oils. Those of which prove essential for operation in tempering plants would then be introduced into practice. Furthermore, the article describes monitoring the changes in the properties of a quenching oil used in a hardening shop, the effects of quenching oil temperature on its cooling capacity and the impact of the water content on certain cooling parameters of selected oils. Cooling curves were measured (including cooling rates and the time to reach relevant temperatures) according to ISO 9950. The hardening power of the oil and the area below the cooling rate curve as a function of temperature (amount of heat removed in the nose region of the Continuous cooling transformation - CCT curve) were calculated. V-values based on the work of Tamura, reflecting the steel type and its CCT curve, were calculated as well. All the data were compared against the hardness and microstructure on a section through a cylinder made of EN C35 steel cooled in the particular oil. Based on the results, criteria are recommended for assessing the suitability of a quenching oil for a specific steel grade and product size. The quenching oils used in the experiment were Houghto Quench C120, Paramo TK 22, Paramo TK 46, CS Noro MO 46 and Durixol W72.

**Keywords:** cooling curves; quenching oil; hardening power; V-value; ISO 9950; steel C35E

## 1. Introduction

This paper responds to the state-of-the-art practice in monitoring the cooling efficiency of quenching oils in industrial practice. Ordinarily, hardening shops only request a quenching process report for a quenching oil. However, it is rather complicated to interpret the values appropriately.

This paper compares various criteria for assessing quenching oils and reviews their relevance to appropriate oil selection. The criteria were compared on the basis of core hardness in an EN C35E steel product. Chemical composition of the steels is on Table 1. Those that are crucial to achieving the desired properties are then chosen. Five fast to medium-speed quenching oils were used in this investigation. They were Paramo TK 22, Paramo TK 46, CS Noro MO 46, Houghto Quench C 120 and Durixol W72. The goal was not to compare these oils as such, but to select a criterion that best describes their hardening power. The following criteria were compared:

- Cooling curve (including the cooling rate and time to cool to certain temperatures) according to ISO 9950. This part of the experiment is based on [1].
- Hardening power. This part of the experiment is based on [2]. The measurement and calculation of hardening power for the oils builds on [3].
- Area under the cooling curve at certain temperatures (the amount of heat removed in the nose region of the Continuous cooling transformation curve). This is based on [4,5].

- V-value was developed by Tamura, and reflects the steel grade or, more appropriately, its CCT curves [6].
- Profile of hardness and microstructure on a section through a cylinder made from EN C35E steel for a particular oil.

**Table 1.** Chemical composition of the steels in weight percent.

| C | Mn | Si | P | S |
|------|------|------|-------|-------|
| 0.36 | 0.6 | 0.22 | 0.011 | 0.015 |

The planned outcome is to determine the single most relevant criterion for assessing various quenching oils.

An important part of this investigation was monitoring the effect of the water content on the hardening power of the quenching oil. This was done for the Houghto Quench C120 oil. The purpose was to determine the response of the oil's hardening power to increased water content. It was measured using an Inconel probe for testing in accordance with ISO 9950. This experiment therefore builds on the D. Scott MacKenzie study [7]. Monitoring the oil ageing process in a hardening shop was inspired by study [7] as well. Over the course of 24 months, the cooling efficiency of the NORO MO 46 oil was monitored in a custom hardening shop, which quenched eight charges a day [8,9].

Finally, the relationship between the quenching bath temperature and hardening power of the oil was monitored. This part of the experiment is based on N. Kobasko's work [10].

## 2. Identification of an Appropriate Criterion for Assessing of Quenching Oils

In this section, the following criteria are compared: hardening power, area under the cooling curve, V-value and the hardness profile of a quenched cylinder. The assessment is based on a hardness profile on a section through a quenched cylinder from EN C35E steel 30 mm in diameter.

### 2.1. Comparison of Quenching Oils Based on Hardness and Microstructure of EN C35E Steel

The criterion for comparing quenching oils was a hardness profile on a section through the cylinder (Table 2, Figure 1). EN C35E steel was chosen for its relatively low hardenability. In this steel, the difference between cooling intensities should be manifested more strongly than in steels with higher hardenabilities. The specimens were cylinders 50 mm height and 30 mm in diameter. Samples were cut in the middle section. In Figure 1, there is a slight difference between Paramo TK 46 oil and the others. The main reason could be the different wetting between Paramo TK 46 and the others.

**Table 2.** Hardness profile mean value and deviation for individual oils.

| Depth [mm] | Oil | | | | |
|---|---|---|---|---|---|
| | CS Noro MO 46 | Paramo TK 46 | Durixol W72 | Paramo TK 22 | Houghto Quench C120 |
| | HV1 Mean and Standard Deviation | | | | |
| 1 | $219 \pm 3$ | $238 \pm 3$ | $223 \pm 4$ | $227 \pm 2$ | $228 \pm 3$ |
| 3 | $217 \pm 3$ | $228 \pm 3$ | $219 \pm 4$ | $222 \pm 3$ | $220 \pm 3$ |
| 5 | $215 \pm 4$ | $226 \pm 4$ | $219 \pm 3$ | $225 \pm 3$ | $215 \pm 3$ |
| 7 | $214 \pm 3$ | $226 \pm 3$ | $213 \pm 2$ | $217 \pm 3$ | $218 \pm 3$ |
| 10 | $207 \pm 4$ | $215 \pm 3$ | $212 \pm 4$ | $218 \pm 2$ | $215 \pm 4$ |

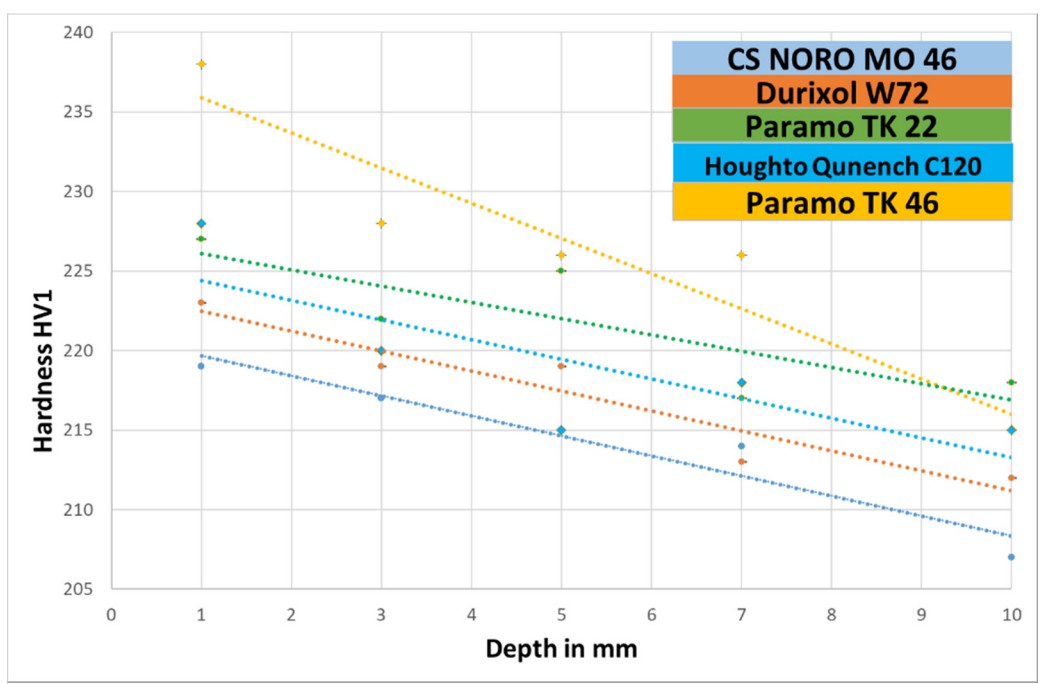

**Figure 1.** Hardness profile for individual oils.

In addition, the microstructure in the surface of the steel was examined. Metallographic sections show that the relatively high cooling rate in Paramo TK 46 oil led to a larger volume fraction of pearlite in the microstructure.

The fractions of phases and their mixtures in the as-quenched microstructure depended on the cooling rate and on the type of oil. There was a ferritic-pearlitic microstructure in the surface of the specimen quenched in CS Noro MO 46 (Figure 2a). At this magnification, large proeutectoid ferrite grains and fine pearlite are observed.

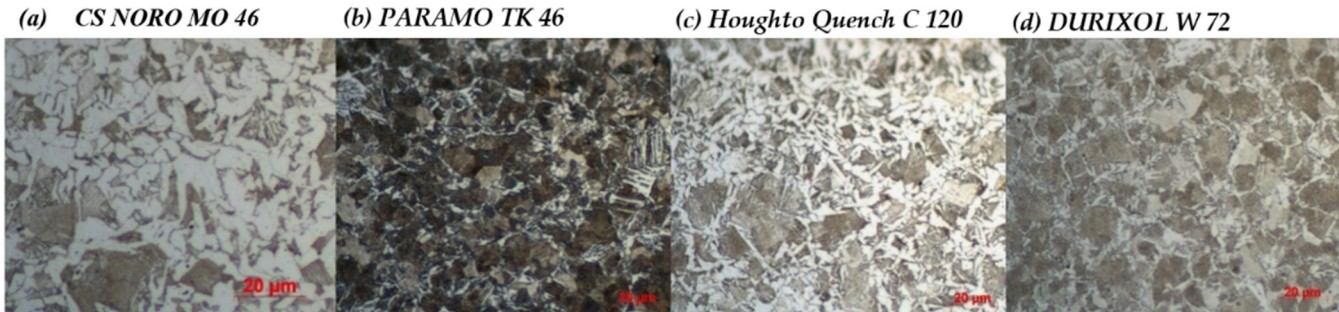

**Figure 2.** Microstructures in C35E steel at 1 mm below the surface. Quenched in oil. From the left: (**a**) CS NORO MO 46, (**b**) PARAMO TK 46, (**c**) Houghto Quench C 120, (**d**) DURIXOL W 72. Magnification 1000×.

The as-quenched microstructure in the specimen quenched in Paramo TK 46 oil consisted of pearlite and ferrite (Figure 2b). It was very fine pearlite and proeutectoid Widmanstätten ferrite. Ferrite was also found along the boundaries of prior austenite grains. The surface microstructure of the Houghto Quench C120 oil consisted of ferrite and pearlite (Figure 2c). A distinct ferrite network can be seen on prior austenite grain boundaries. Pearlite was very fine with scattered large ferrite grains and acicular ferrite. Quenching in Durixol W72 produced a pearlitic-ferritic microstructure (Figure 2d). Ferrite is only present in the form of a network and is practically free from large grains. In this case, the pearlite is very fine.

## 2.2. Cooling Curves According to ISO 9950

The cooling curves were measured and compared for the same set of oils as in the previous section. Measurement was performed according to ISO 9950. The temperature of the oils was 50 °C (Figure 3).

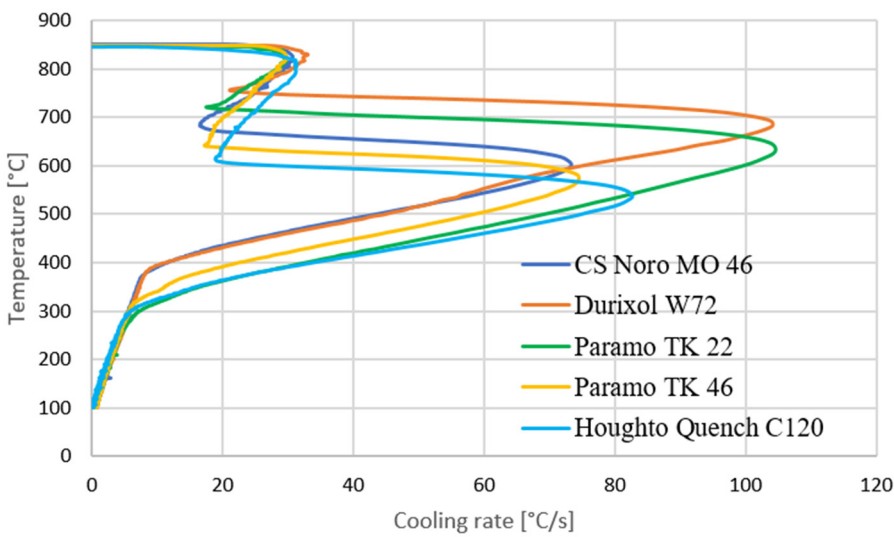

**Figure 3.** Cooling curves (temperature—cooling rate) according to ISO 9950.

The data clearly show that all the oils are medium-speed to fast oils. In addition, they are what are called "cold oils," which best perform at 50–70 °C. The fastest-cooling oils are Paramo TK 22 and Durixol W72, whose maximum cooling rates exceed 100 °C/s. For both of these very fast oils, the $CR_{max}$ is above 600 °C, which is an acceptable range for unalloyed steels (EN C45, C35). The nose of the CCT diagram is approximately at 550 °C [11]. This condition is fully met by Houghto Quench C120 oil. The shortest stable vapor blanket stage was found with Durixol W72. This oil also has the longest convection stage (together with CS Noro MO 46). All the calculated values for individual oils are listed in Table 3.

**Table 3.** Basic parameters derived from the cooling curve.

| Oil | $CR_{max}$ [°C/s] | Temperature When Reaching $CR_{max}$ [°C] |
|---|---|---|
| CS Noro Mo 46 | 73 | 603 |
| Durixol W72 | 104 | 681 |
| Paramo TK 22 | 104.5 | 636 |
| Paramo TK 46 | 74 | 574 |
| Houghto Quench C120 | 83 | 537 |

## 2.3. Hardening Power

Hardening power (HP) can be used as a criterion for selecting an appropriate quenching oil. The method involves finding three transition parameters from the cooling rate for the relevant quenching oil and substituting them in the equation based on C45 steel: [1,12].

$HP = 91.5 + 1.34\, T_{VP} + 10.88\, CR - 3.85\, T_{CP}$, where

$T_{VP}$ ... Temperature of the transition between the vapor blanket and the boiling stages,

$CR$ ... Cooling rate at 550 °C and

$T_{CP}$ ... Temperature of the transition between the boiling stage and the convection stage.

The highest hardening power was found for Paramo TK 22 (Figure 4). All the calculated values for individual oils are listed in Table 4.

$$HP = 91.5 + 1.34 \times T_{vp} + 10.88 \times CR - 3.85 \times T_{cp} =$$
$$= 91.5 + 1.34 \times 720 + 10.88 \times 84.45601 - 3.85 \times 288 = 866$$

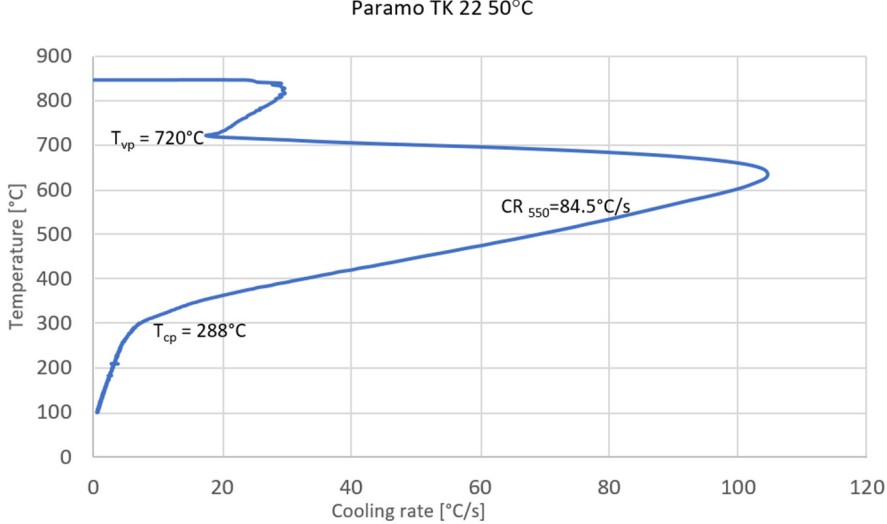

**Figure 4.** Paramo TK 22 parameters found from the cooling curve.

**Table 4.** Hardening power for individual oils.

| Oil | Hardening Power |
|---|---|
| Houghto Quench C120 | 659 |
| CS Noro MO 46 | 229 |
| Durixol W72 | 260 |
| Paramo TK 22 | 866 |
| Paramo TK 46 | 515 |

To validate the HP values, they were correlated with the hardness value measured at 10 mm below the surface, as outlined in the previous section (Figure 5).

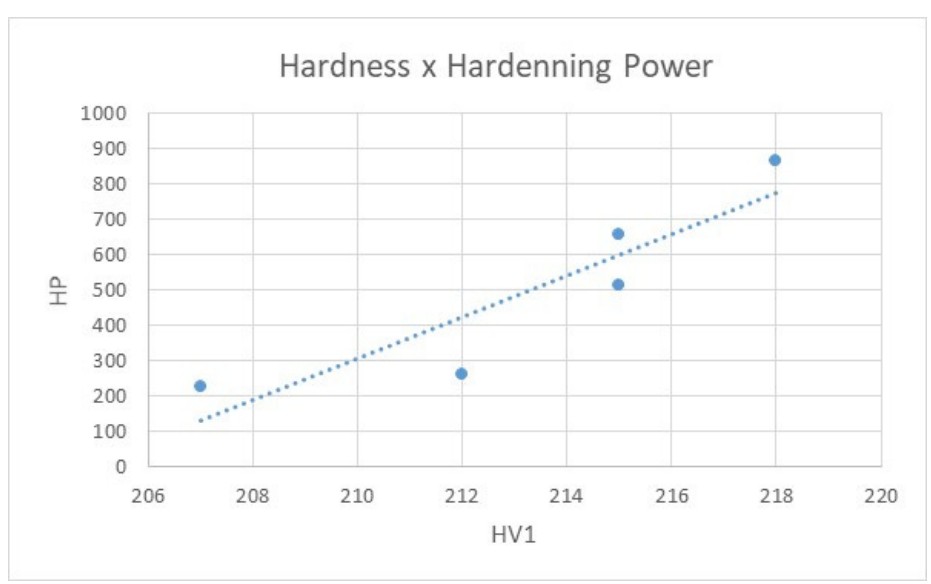

**Figure 5.** Correlation between hardness and Hardening power.

There is a clear, directly-proportional relationship between hardening power and the core hardness of the quenched C35 steel. The best-performing oil was found to be Paramo TK 22. The above equation was developed for calculating the quenching power for C45 steel, but it is widely used and universal. The key term in this equation is the contribution

from the maximum cooling rate. As there was excellent agreement between hardness and HP value, this criterion appears to be appropriate for characterizing quenching oils.

### 2.4. Assessment Based on the Area under the Cooling Curve Depending on Temperature

This method involves integrating the cooling rate function to obtain the area under its curve. The goal is to find a single criterion that characterizes the cooling process (the correlation between the cooling rate and as-quenched hardness). However, it is very difficult to predict the quenching process using a single parameter of the cooling curve. For this reason, it is useful to model the heat transfer process for the entire quenching period. Božidar Liščić has shown that one way to obtain a correlation between the quenching rate and hardness is to integrate the cooling curve [3,13]. A similar method was proposed by K. E. Thelning, who showed that integration yields the area under the cooling curve between two critical temperatures $T_1$ and $T_2$. These critical temperatures depend on the chemical composition of the steel. Using this method, it was found that the area under the curve increases with the mean quenching rate. K. E. Thelning also showed that the optimal temperature range depends on chemical composition [4,8]. In the present case, the area is bound by temperatures of 600 °C and 300 °C. The cooling curve was integrated using Wolfram Mathematica software (Figure 6). The graphs in this software were generating using Wolfram Language. With the aid of Wolfram Mathematica, cooling curve graphs were plotted for the 300–600 °C range. The legend of the graph includes the integral values for areas under the curve for various oil temperatures.

The area under the curve indicates the amount of heat removed from the machine part. Most laboratories determine the area between 600 °C and 300 °C. The most heat was removed by Paramo TK 22, and then Houghto Quench C120. By contrast, the least heat was removed by Durixol W72 and CS Noro MO 46 oils. What is clear to see is the correlation between the amount of heat removed and the core hardness of the cylinder from C35E steel. This criterion also highlights the differences between quenching baths at various temperatures. The quenchant's cooling efficiency decreases with its increasing temperature. Durixol W72 and CS Noro MO 46 are exceptions as their hardening power expressed as the area under the curve does not vary.

### 2.5. V-Value

In an effort to simplify the interpretation of the cooling curves, I. Tamura introduced a criterion referred to as the V-value. It is intended to characterize the ability of a quenchant to harden steel. The V-value is defined as the ratio of the boiling stage temperature interval on an oil's cooling curve and the temperature interval for steel transformation for which the fastest cooling rate is required. It can be calculated from cooling curve data and the corresponding CCT diagram: [6].

Tamura also established a classification system for steels, which he divided into four categories dictated by the shape of their CCT diagrams. For each of the categories, he adjusted the basic temperature interval ratio, as indicated above. The following representatives were selected for the categories: EN C45, EN C80, EN 100Cr6 and EN 25CrMo4. For each of them, a CCT diagram was developed with a cooling curve, as illustrated below.

The following temperatures were derived from the graph in Figure 7: $T_c$ = 639 °C, $T_s$ = 657 °C, $T_f$ = 523 °C and $T_d$ = 317 °C.

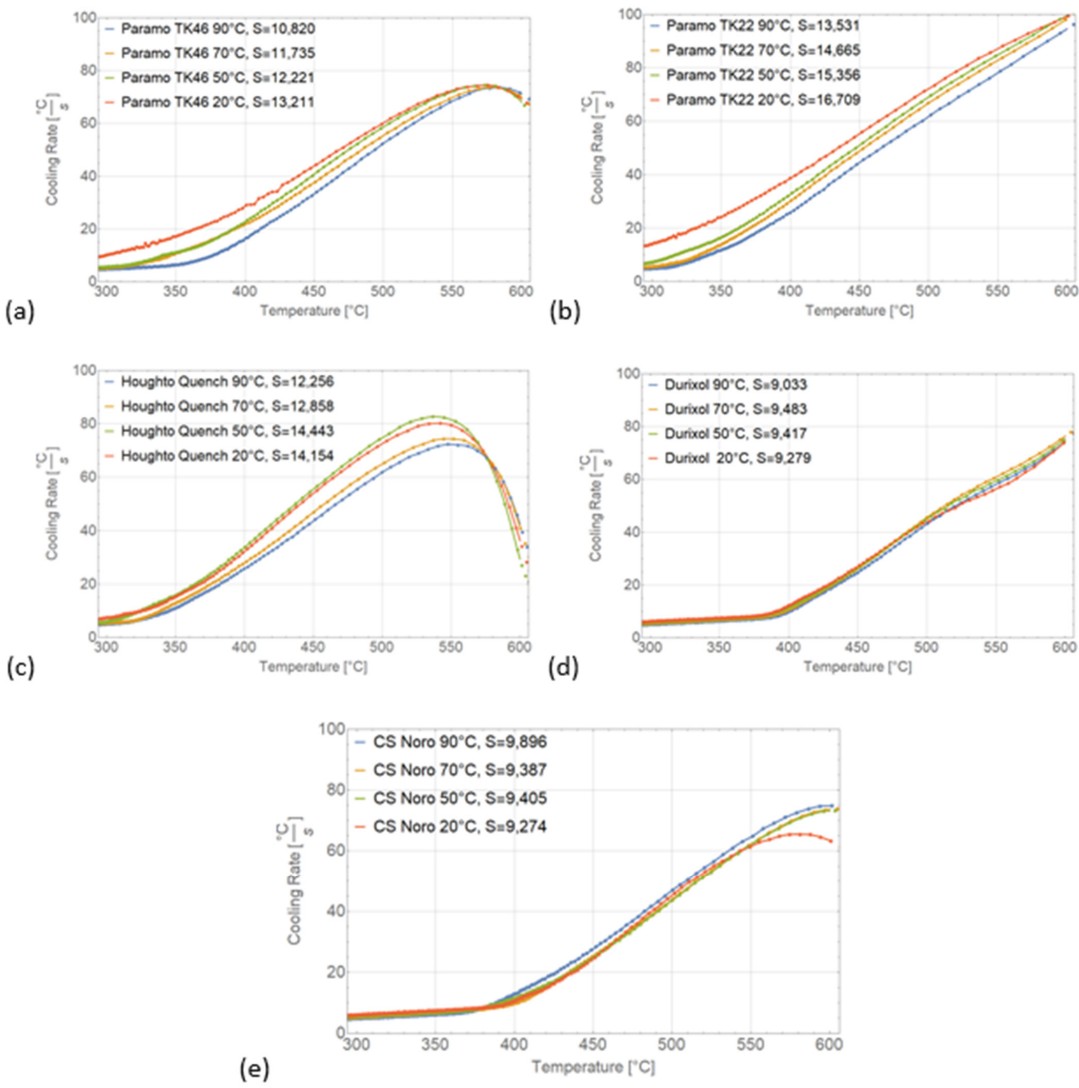

**Figure 6.** Results of integration yield the areas under the curve for cooling between 300–600 °C at the quenchant temperatures of 20 °C, 50° C, 70 °C and 90 °C. (**a**) Paramo TK46, (**b**) Paramo TK 22, (**c**) Houghto Quench, (**d**) Durixol, (**e**) CS Noro.

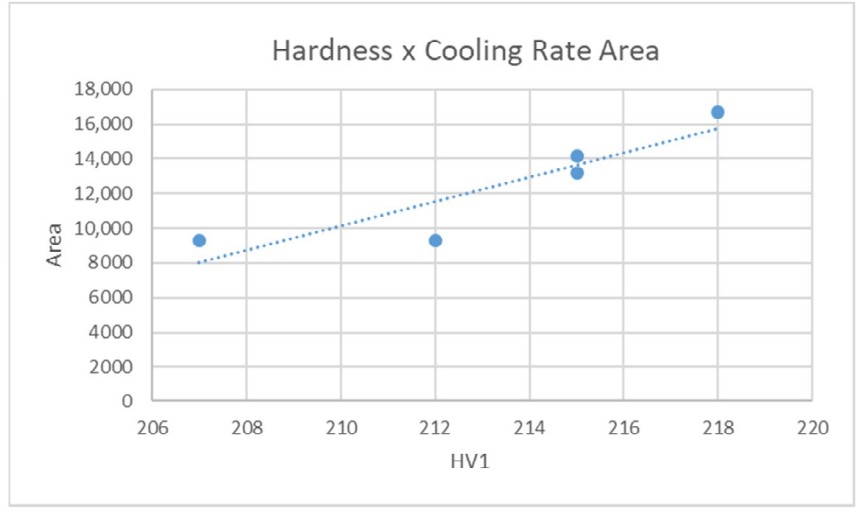

**Figure 7.** Hardness vs. amount of heat removed.

Here, $T_c$ is the temperature at the start of the boiling stage, $T_d$ is the end of the boiling stage, $T_s$ is the temperature at the start of the transformation to "soft phases," i.e., austenite—ferrite or pearlite, and $T_f$ is the temperature at the start of the austenite-martensite transformation (Figure 8). For this steel category, Tamura reports:

$$V = \frac{T_c - T_d}{T_s - T_f}, \text{ and after substitution, the V-value was calculated as: } V = \frac{T_c - T_f}{X} =$$
$$\frac{T_c - T_f}{T_s - T_f} = \frac{639 - 523}{657 - 523} = 0.866.$$

The Table 5 gives a summary of the results:

**Table 5.** Calculated *V*-values for relevant steels and oils.

| Oil | Steel | | | |
|---|---|---|---|---|
| | **C45** | **C80** | **100Cr6** | **25CrMo4** |
| CS Noro MO 46 | 1 | 1 | 0.98 | 0.96 |
| Durixol W72 | 1 | 0.8 | 0.81 | 0.86 |
| Paramo TK 22 | 1 | 0.73 | 0.77 | 1 |
| Paramo TK 46 | 0.87 | 1 | 1 | 1 |
| Houghto Quench C120 | 0.68 | 0.74 | 1 | 1 |

The reference parameter was the one for EN C45 steel whose characteristics are very close to those of EN C35 steel. Clearly, this parameter is not in agreement with the hardness obtained after hardening (Figure 9).

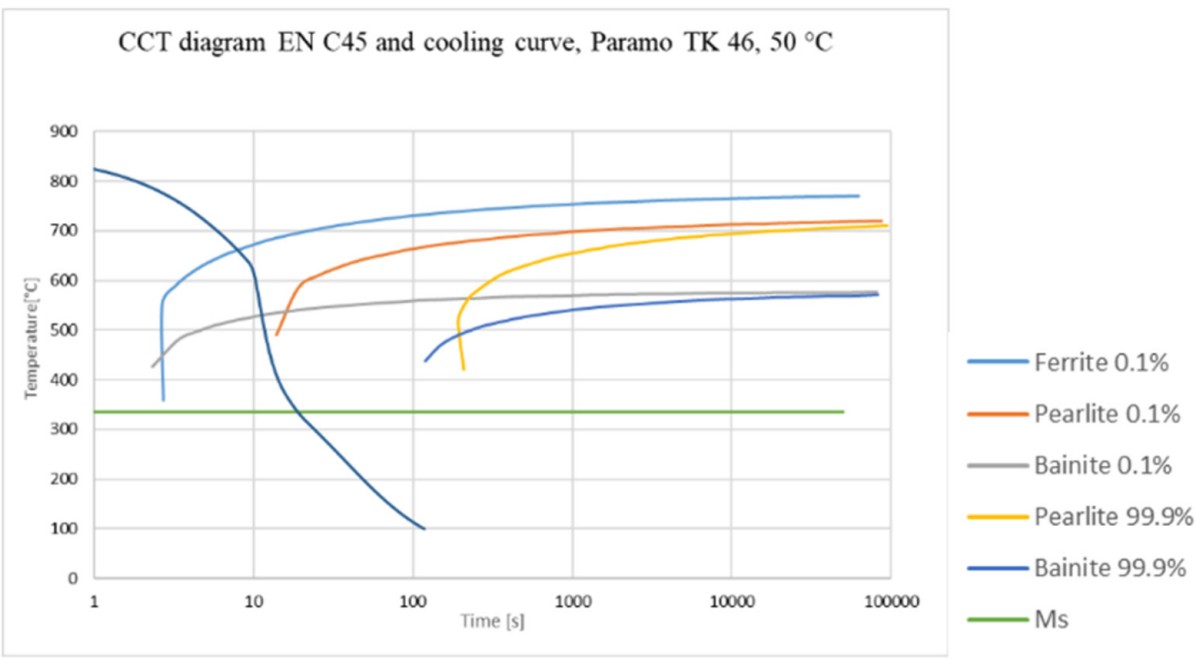

**Figure 8.** Cooling curve for Paramo TK 46 oil (measured at 50 °C) plotted in a CCT diagram of C45 steel.

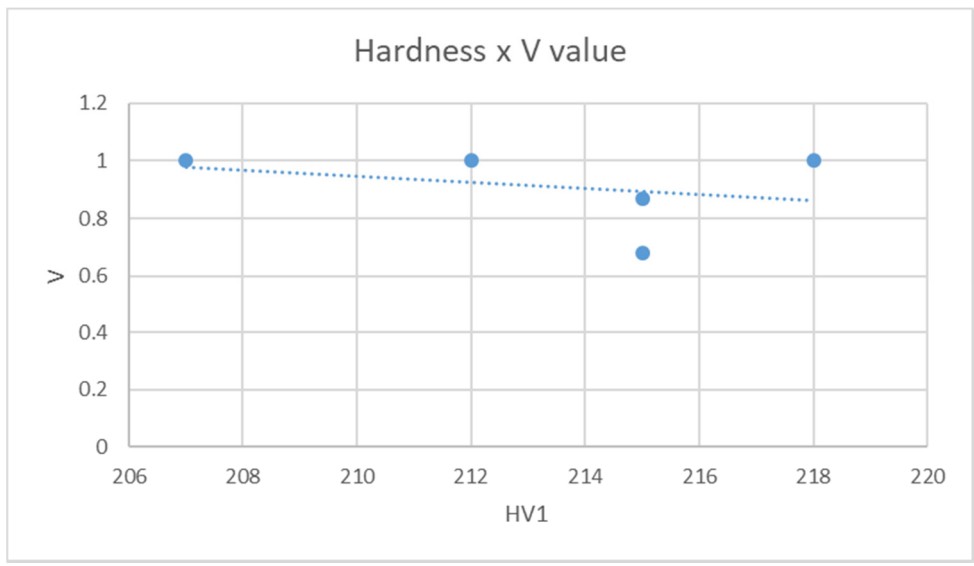

**Figure 9.** Correlation between hardness and V-value.

## 3. Effect of Water Content on Cooling Efficiency of Selected Oils

Water content in quenching oil is one of the most frequently-discussed issue in the day-to-day operation of a hardening plant. In our experiment, the following amounts of water in oil were considered: 0%, 0.01%, 0.1%, 0.5% and 1% (Figure 10). The effect of water content on temperature-time and temperature-cooling curves was measured in Houghto Quench C120 oil. These curves were measured for water contents of 0%, 0.01%, 0.1%, 0.5% and 1%. For safety reasons, the measurement was not carried out for any water content above 1%. In line with ISO 9950, the temperature during measurement was 40 °C. The measured data only reveal a minimum impact of higher water levels on the stable vapor blanket stage. The CRmax for this oil decreases slightly with changes in water content. The temperature that corresponds to CRmax does not change. The start of the convection stage is shifted slightly towards higher temperatures.

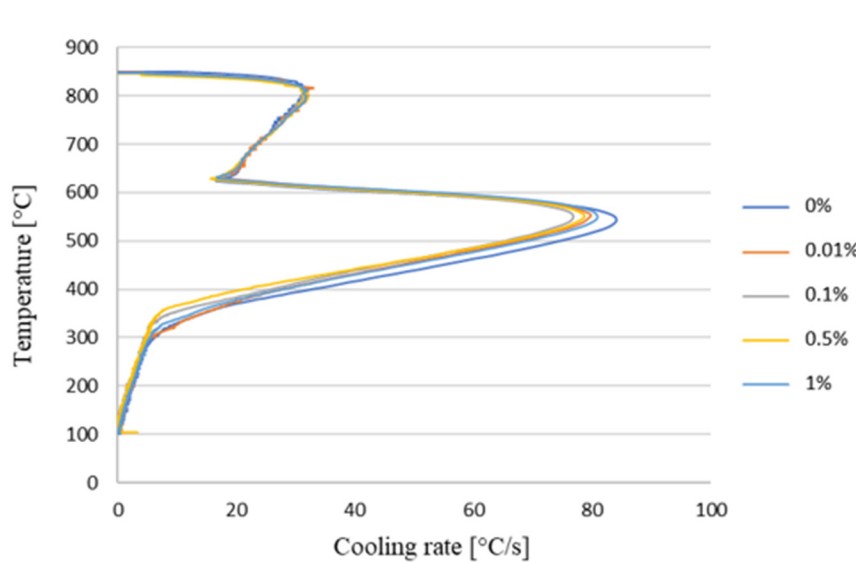

**Figure 10.** The effect of water level in Houghto Quench C120 oil on its hardening power (temperature-cooling curve).

## 4. Monitoring Ageing Oil in a Hardening Shop

Over the course of almost 24 months, specimens of a medium-viscosity quenching oil were taken in a custom hardening shop and their hardening characteristics were measured. The purpose of this experiment was to determine whether the volume of oil added to replenish the quenching tank is sufficient for maintaining the desired quenching characteristics. The progress of ageing was measured for CS Noro MO 46 oil by evaluating its curves. A total of 10 specimens of CS Noro MO 46 oil were measured (Figure 11).

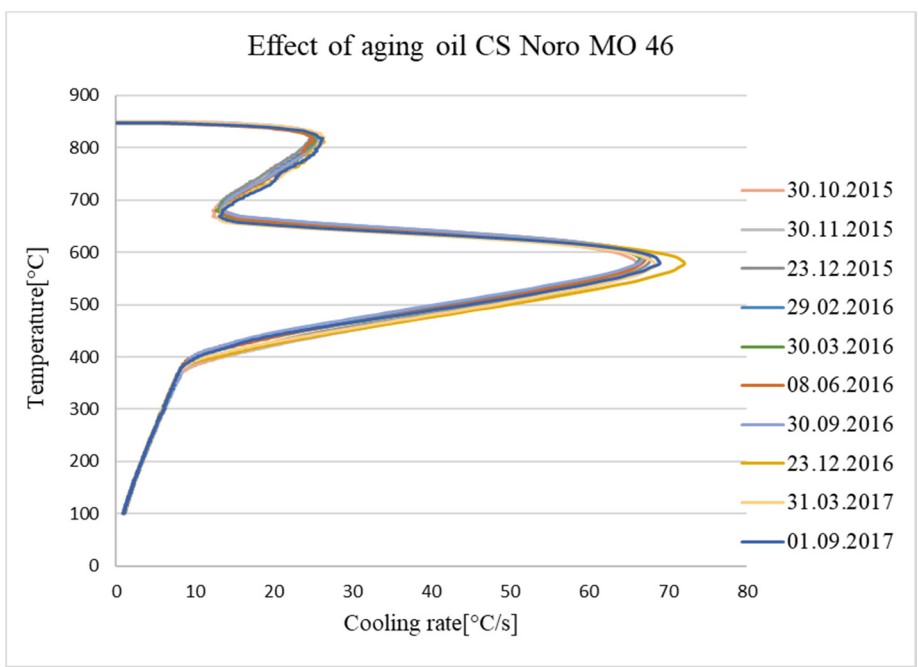

**Figure 11.** Ageing of CS Noro MO 46 oil.

The plot shows that the change in the temperature interval in which the vapor blanket remains stable is minimal between the individual runs. Fluctuations in the course of service are only found for the CRmax value. By contrast, the temperature at which CRmax was reached remained virtually unchanged. This finding is highly relevant, given the desired product hardness levels. The starting point of the convection stage did not change either.

## 5. Measurement of Flash Point and Fire Point of Quenching Oils

The quenching oils were experimentally tested according to EN ISO 2592. The fire point of a quenching oil is essentially the most important characteristics with respect to safety and potential consequences of a fire in a hardening shop. The outcome of this experiment was an overview of these temperatures depending on the quenching oil type and hardness (Table 6).

**Table 6.** Flash points for a quenching oil, hydraulic oil, engine oil and a synthetic oil.

| Oil | Flash Point [°C] |
|---|---|
| CS Noro MO 46 | 220 |
| Paramo TK 22 | 191 |
| Paramo TK 46 | 216 |
| Durixol W72 | 216 |
| Houghto Quench C120 | 196 |

When we compare the flash point values and as-quenched hardness levels, it becomes clear that higher hardness is obtained in oils with lower flash points. It indicates a

straightforward relationship between the amounts of oil fractions and additives and the key properties of quenching oils.

## 6. Discussion

### 6.1. As-Quenched Hardness

The choice of hardness was motivated by the need to quantify the hardening power of oil, which is relevant to industrial practice. The criterion for assessing cooling efficiency was the hardness of the quenched specimens 10 mm from their end. The results clearly show that the optimal choice would be a steel with higher hardenability, such as 42CrMo4, in the form of specimens with larger diameters (50 mm). One drawback of EN 42CrMo4 steel is the fact that the hardening power (HP) criterion is calculated for carbon steels (with carbon levels of 0.3–0.55%), and therefore, the values would require correction. All the comparisons were based on a simple underlying principle: the higher the core hardness of the steel cylinder, the higher the core hardening power of the oil.

### 6.2. Cooling Curves of Quenching Oils

The cooling curves suggest that all the oils are medium-speed to fast oils. Individual vapor blanket stage intervals and vapor separation can be seen in the curves. The highest cooling rates are obtained with Paramo TK 22 and Durixol W72, whose maximum cooling rates exceeded 100 °C/s (104.5 °C/s and 104 °C/s, respectively). In both cases, this speed was achieved at temperatures above 600 °C (636 °C and 681 °C, respectively).

### 6.3. Hardening Power

Hardening power (HP) and core hardness in C35 steel specimens are directly proportional. The best-performing steel is with Paramo TK 22, which showed the highest HP (866). The second highest HP (656) was found in Houghto Quench C120. The lowest HP (229) was obtained with CS Noro MO 46. As discussed above, there is a directly proportional relationship between hardening power and the hardness of the quenched specimens of EN C35 steel. The highest HP values correspond to the highest maximum cooling rates.

### 6.4. Area under the Cooling Curve

By evaluating this parameter, one can obtain information on the amount of heat removed from a machine part. There is a correlation between the amount of heat removed and the core hardness of the quenched cylinders from EN C35E steel. By evaluating this criterion, one can secure the best tool for monitoring even slight changes in hardening characteristics, such as the effect of temperature on the hardening power of the oil. For these reasons, it is the most relevant criterion and can be recommended as the criterion of choice for expressing hardening power. The right selection of a temperature interval for evaluating the area under the curve is a crucial factor. This temperature interval should be chosen with respect to the CCT diagram of the steel. The measurement revealed that the largest amount of heat was removed by Paramo TK 22 at 20 °C ($16\,709\ s^{-1}$). By contrast, the lowest value was found with Durixol W72 at 90 °C ($9033\ s^{-1}$) and CS Noro MO 46 at a temperature of 20 °C ($9274\ s^{-1}$).

### 6.5. V-Value

The data from this measurement do not correspond to the hardness values measured. The difficulty with this method is that it is not versatile enough. In order to obtain valid information, the measurement procedure must be adjusted for the steel grade (specimens of a particular chemical composition) and the specimen geometry. In addition, each specimen should be fitted with thermocouples. Values collected in this manner would, understandably, have more validity. Another weakness of this method is the rather complicated determination of temperatures for substituting in the formula.

## 7. Conclusions

- The most relevant criterion for the hardening power of an oil is the area under the cooling curve, which characterizes its ability to remove heat from a part. The criterion must be evaluated within a temperature interval governed by the type of steel. The area between 600 °C and 300 °C is the most versatile option that is appropriate for the majority of structural steels.
- The HP criterion is useful for comparing medium-speed and fast oils. HP is less sensitive to changes in quenching characteristics than the area under the curve.
- The V-value appears to be the least suitable for this purpose, mainly because it is complicated to evaluate. In addition, one must always tailor the experiment to the materials and products to be quenched.
- In our experiment, the main criterion was the core hardness of a steel cylinder from EN C35E. Specimens with a larger diameter and a steel with higher hardenability would be a better choice. The authors recommend a cylinder of 50 mm diameter from EN 42CrMo4 steel.

**Author Contributions:** Writing—original draft preparation, Z.D.; writing—review and editing, J.H.; funding acquisition, J.H.; project administration, V.P. All authors have read and agreed to the published version of the manuscript.

**Funding:** "This research was funded by ERDF, grant number CZ02.1.01/0.0/0.0/16_019/0000836" and by specific university research SGS—2018-051".

**Informed Consent Statement:** Informed consent was obtained from all subjects involved in the study.

**Conflicts of Interest:** The authors declare no conflict of interest.

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
