# Peer review of "Comparison of Industrial Quenching Oils"

_metals, doi:10.3390/met11020250_

Round 1

Reviewer 1 Report

In this paper, selected criteria for assessing of quenching oils and reviews their relevance to appropriate oil selection were compared on the basis of core hardness in a steel product. Those which are crucial to achieving the desired properties should then be chosen. Five fast to medium-speed quenching oils were used for this investigation. From my point of view, this paper is considered to be a technical paper. However, it contains valuable information, we can see it in a catalog or an instructions booklet for. Especially, it suitable for a specific category of beneficiaries.

Author Response

Dear Reviewer,

The present paper has been written for a special issue of Metals which focuses on heat treatment. Quenching media and their characteristics are a fundamental issue in heat treatment. The criteria chosen here are relatively complex and are defined on an unambiguous and scientific basis. Complex definitions of the criteria and their interpretations have been presented in a number of research publications in renowned journals. The problem addressed by the present paper is the profound lack of adequate comparison among the criteria. This involves by no means the comparison between oils as such. Until today, no publication has dealt with comparison of this kind. Therefore we believe that it is important to publish this study in this special issue on heat treatment. Wording and graphic layout have been updated and arithmetic mean and variance values have been added, among other modifications.

Best regards

Author. 

Reviewer 2 Report

Dear author,

I have some comments to your interesting paper.

  • Please check you English, e.g. passive, plural vs. singular, BE vs. AE, etc.
  • Where did you cut the cylinder? It is not clear from which position of the cylincer you'd machined the specimens. Are those from the ends ore from the middle section?
  • Could you provide the chemical composition of the steel. This makes it easier for the reader to follow.
  • For better comparison you should use the same colors for each oil in the figures, e.g. Paramo 46 is dark orange in fig. 1 but yellow in fig. 3.
  • Can you comment on the drop of Paramo 46 in fig. 1? The other oils have a similar effect on hardness.
  • I would suggest a different color for the socond blue curve, e.g. green. The different shades of blue are hard to distinguish.
  • Can you provide the data from which you concluded the nose in fig. 3 is at about 550°C?
  • Paramo 22 and Durixol are fast cooling oils but their hardening power differs by more than 600. Can you commnet on that? I can follow the equation for HP and see that this delta is based in the delta T in the when reaching CR max but is it reasonable to compare such different oils?
  • According to fig. 4 I would say that Durixol does not fit. This is the same for fig. 6.
  • You'd measured the hardness 10 mm from the surface, i.e. along the circumference. Did you also measure a hardness profile through the whole diameter of the cyclinders?
  • You should add directly at the beginning of chapter 1.3 that the equation is based on C45 steel.
  • The y-axis in fig. 5 should have the same range for all oils.
  • What is what in fig. 7?
  • Fig. 8 differs form figs. 4, 6. In this curve the value of Durixol fits better.
  • Can you provide a curve for all oils with an equal water content?
  • You recommend 42CrMo4, why? This alloy has a higher Cr content than C45 or C35.
  • You said that the core hardness is a critical value, but you did not measure it! You measured the hardness 10 mm in depth of a cyclinder 30 mm in diameter.
  • What are the mean values for the hardness. I guess you'd made several measurements. The deviation would be also a key value to know. 
  • Best regards
  • Reviewer

Author Response

Dear Reviewer,

All comments are in the attachment.

Best regards

Author. 

Round 2

Reviewer 1 Report

The paper can be accepted in the current form for the special issue 

Author Response

Dear reviewer,

Thank you for your review. English style was corrected. 

Best regards,

Jiri